# When do science recommendations stop being effective? The case of the sprawl of diesel electricity generators in Beirut

**Mohammad Al Hallak**[iD], **Abdul Aziz Kurdieh**[¤a], **Amira Yassine**[¤b], **Rachel El Hage**, **Najat Saliba**[iD]*

Department of Chemistry, American University of Beirut, Beirut, Lebanon

¤a Current address: Laboratory of Atmospheric Chemistry, Paul Scherrer Institute (PSI), Villigen, Switzerland
¤b Current address: Environmental Health and Engineering Department, Johns Hopkins University, Baltimore, Maryland, United States of America
* najataoun.mp@gmail.com, ns30@aub.edu.lb

**Data Availability Statement:** All relevant data are within the paper and its Supporting information files.

## Abstract

Lebanon, plagued by political and economic crises, experienced a government collapse in early 2020, leading to an electrical nationwide blackout by 2023. Diesel generators were used to compensate for the absence of power production from the national provider, Electricité du Liban (EDL). To investigate the effect of the crisis on the levels of 16 EPA particle bound polycyclic aromatic hydrocarbons (PPAHs), an annual comparative analysis of three locations within Beirut started in 2022 and ended in 2023. The locations are: American University of Beirut (AUB), Beirut Central District (BCD) and Nursing School Makassed University (NSMU). Sampling took place and the PPAHs samples were extracted, quantified using Gas Chromatography-Mass spectrometry (GC-MS) and source apportioned using Positive Matrix Factorization (PMF). Particulate Matter 2.5μm ($PM_{2.5}$) mean levels at AUB, BCD and NSMU, which was found to be 14.3 μg/m$^3$, 18.3 μg/m$^3$ and 22.6 μg/m$^3$ respectively, beside the high annual $PM_{2.5}$ mean level (17.19 μg/m$^3$) exceeded the World Health Organization (WHO) standard levels. The factors identified in the three sites are diesel, incineration, and gasoline emissions. The dominant factor in three sites was the diesel emissions, specifically from generators, with 56% in BCD, 42% in AUB and 43% in NSMU. The contribution of diesel emission in AUB has increased by 100% since the last study in 2016–2017. Similarly, the excess cancer risk (ESR) in the three sites was above the EPA threshold with an increase of 53% compared to the one calculated previously in AUB. This situation, where law of enforcement is absent, need for international action to curb emissions and for funding agencies to adopt sustainable, "carbon-free" funding strategies to support urban development in low- and middle-income countries (LMICs). Yet, EDL's failure to fulfill Lebanon's populace electricity requirements infringes upon their electricity entitlements.

**Funding:** The author(s) received no specific funding for this work.

**Competing interests:** The authors have declared that no competing interests exist.

## Introduction

Lebanon, beset by political instability and economic woes, witnessed a governmental collapse in early 2020. Mounting debt, currency devaluation, and soaring unemployment exacerbated social unrest and economic turmoil [1]. The collapse led to a complete shutdown of the national electricity provider, Electricité du Liban (EDL), resulting in widespread blackouts starting 2021. These blackouts were compensated for by diesel generators at both the building and neighborhood levels [2].

The Electricité du Liban (EDL) was already in a fragile financial situation before the collapse. As Lebanon's sole, vertically integrated utility responsible for the provision of electricity nationwide, EDL has been plagued by mismanagement. The dire state of EDL's service provision has led to the proliferation of private diesel generators [3–6]. According to a recent World Bank study, diesel generators contribute 8.1 terawatt-hours (~37 percent) of the country's total electricity demand [7]. Over the years, the influence of diesel generator network owners has grown, partly due to the emergence of powerful neighborhood-level structures that have significant implications for energy access [8]. Until recently, the operations of these networks were neither regulated nor monitored.

Since 2010, the proliferation of diesel generators has reached unprecedented levels, with Beirut alone accommodating over 9300 of these units by 2017 in an area of 20 km$^2$ [9]. Initially, electricity rationing lasted three hours daily in 2010, but by 2022 and 2023, it escalated to over 20 hours. Assembled domestically, these generators emit significant greenhouse gases and carcinogenic particles, alongside high levels of reactive oxygen species. Monitoring data on these emissions since 2010 reveals a concerning trend, with the presence of carcinogens on the rise [10].

In 2010, Shihadeh and colleagues examined the levels of airborne carcinogens on the balconies of 20 residences in Beirut's Hamra area, comparing periods when diesel generators were operating to when they were off. They discovered that using diesel generators for just three hours daily contributed to 38% of the daily carcinogen exposure in areas confined between buildings. This represents a roughly 60% increase over the background levels without generator use [11]. The added carcinogen exposure from diesel generators was comparable to smoking at least two cigarettes per day and would be significantly higher in areas with more extended power cuts [12].

PPAHs are an environmental concern specifically in countries dealing with economic crises. PPAHs can have detrimental effects on environment and human health. The economic crises can impact energy policies, leading to increased reliance on cheaper but environmentally harmful energy sources like diesel generators, wood and biomass burnings and waste incinerations.

Taking AUB as an urban background site, measurements of the levels of particle-bound polycyclic aromatic hydrocarbons (PPAHs) in 2012 over a two-month period in the summer were below the detection limits [13, 14]. In 2015, Saliba and coworkers measured PPAHs at three coastal locations in Lebanon, including AUB and showed that the total PPAHs levels were quantified at 24.7 ng/m$^3$, with Benzo[a]pyrene (BaP), a type 1A carcinogen according to IARC-WHO, measured at 0.49 ± 0.26 ng/m$^3$ in PM$_{10}$ [15]. In 2016–2017, the total PPAHs levels were quantified at 10 ng/m$^3$, and BaP at 0.66 ± 0.05 ng/m$^3$ in PM$_{10}$. Three major sources were identified as contributing to PAH emissions at this urban site: traffic (48%), diesel generators (23%), and incineration (29%) [16]. By 2021, the concentrations of PPAHs remained within the range determined between 2015 and 2019 [17].

The link between PAH prevalence due to diesel generators and obstructive coronary artery disease (CAD) was investigated using a novel marker from coronary catheterization and PAH

markers in a cohort of 258 patients at the American University of Beirut Medical Center between 2014 and 2019. Four types of hydroxylated polycyclic aromatic hydrocarbons (OHPAHs)—2-OHNAP, 2-OHFLU, 3-OHPHE, and 1-OHPYR—were measured in urine samples using high-performance liquid chromatography with a fluorescence detector. The study found higher OHPAH concentrations than those reported in high-income countries, with non-smokers in this study showing higher levels than smokers and some occupational workers elsewhere. This indicates significant exposure to combustion-related PAHs. Notably, 1-OHPYR showed a significant association with obstructive CAD, even after adjusting for variables like age, sex, and diabetes. This highlights the public health implications and the urgent need for regulations to reduce PAH emissions from sources such as vehicles, diesel generators, and incinerators [18].

European countries also faced an increase of pollutants levels during the economic crisis started in 2009 (Eurozone crisis). The increased prices of fuel oil, used as source of energy for domestic heating, encouraged people to burn wood and biomass instead during cold seasons. In Thessaloniki, a greek city, $PM_{2.5}$ and total PPAHs levels increased 30% and 400% respectively in the period where the air was impacted by wood burning emissions [19]. Similarly, Chiang Mai, in Thailand, faced high $PM_{2.5}$ and total PPAHs values reached 65.3±17.5 μg/m$^3$ and 10.23±2.49 ng/m$^3$ respectively during smoke haze period [20].

During total blackouts, people are left with two difficult choices: either live in darkness for few hours or depend on a few solar panels installed on their balconies, if they are fortunate enough to have the space. The lack of a reliable national power supply has driven citizens to adopt these makeshift solutions, highlighting the urgent need for a sustainable resolution to Lebanon's worsening energy crisis.

Given the increasing reliance on diesel generators, now exceeding 20 hours per day due to the near-total collapse of Electricité du Liban (EDL), this study underscores the adverse effects of governmental inaction on emissions. It provides strong evidence of the increased danger these emissions pose to local residents, raising awareness and putting pressure on local officials to take immediate action to reform the electricity sector.

## Methodologies

### Study area and sampling sites

The sampling campaign was conducted in the greater Beirut area at three sites as shown in (Fig 1). The first site was at the American University of Beirut (AUB), serving as an urban background site. This monitoring site, which overlooks the Mediterranean coast from the north/west, is located at the rooftop of the chemistry department (33o 54' 4" N, 35o 28' 45" E, 20m above the ground), and surrounded from all sides by dense vegetation and by a distant that accounts 130 meters far from the highway that get crowded by traffic. The second site was in the Beirut Central District (BCD), also known as Downtown Beirut (DT), located on the rooftop of building 157 (33° 54' 54" N, 35° 30' 20" E, 16.5 meters above ground level). This area, situated on the city's northern coast, is described as the "vibrant financial and commercial hub of the country" and is characterized by the prevalence of diesel generators [2]. Additionally, BCD is still considered a business urban location. It is situated in a pedestrian-only area and the nearest highway is approximately 250 meters away from the site. The third sampling site was on the rooftop of the nursing school at Makassed University (NSMU) (33° 52' 31" N, 35° 30' 12" E, 15 meters above ground level). This location is influenced by emissions from diesel generators and vehicular traffic due to its proximity by 10 meters to the principal roadway with heavy cars loads.

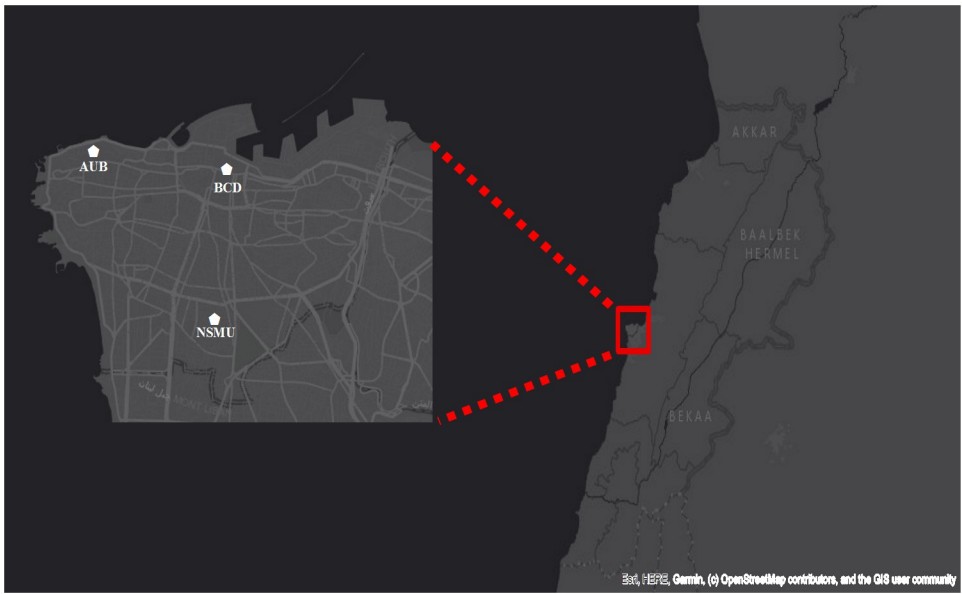

**Fig 1. The three sampling sites in Beirut, Lebanon.**

## Sample collection

Given the widespread presence of diesel generators across most city neighborhoods, the sampling strategy described in the manuscript was designed to capture data from three distinct types of neighborhoods: 1) a high socio-economic status area and business district with a demand for large, high-powered generators, 2) a densely populated, lower socio-economic status area with heavy traffic, and 3) a university site in the city surrounded by greenery. Sampling was conducted at these three locations throughout an entire year to account for temporal variations in pollutant levels at specific sites where generators are frequently in use. By collecting 24-hour samples once every six days, we aimed to create a comprehensive dataset that reflects the fluctuations in pollutant emissions from generators over time.

The campaign involved collecting samples over 24 hours, once every six days. It started in March 2022 and ended in July 2023. Sampling in BCD was conducted from March 2022 to May 2022 and from September 2022 to October 2022. Sampling at AUB took place from June 2022 to August 2022 and from December 2022 to March 2023. Finally, sampling at NSMU occurred from April 2023 to July 2023.

**Collection and gravimetric analysis of PM$_{2.5}$.** A low volume sampler (LVS), composed of a pump and a Harvard cartridge (CHEMCOMB 3500) equipped with PM$_{2.5}$ impactors, operated at a constant flow rate of 16.7 L/min for 24 hours. The samples were collected using 47 mm diameter Teflon PTFE (polytetrafluoroethylene) membrane filters (TISCH Scientific). Gravimetric analysis of PM$_{2.5}$ is performed by measuring the mass difference between pre and post sampling teflon membrane filters using a micro balance (Balance XPR2U) following equilibration under controlled humidity and temperature conditions (40–50% and 20–22°C, respectively). PM$_{2.5}$ levels are reported in μg/m$^3$.

**Collection and chemical analysis of particles bound polycyclic aromatic hydrocarbons (PPAHs).** Samples of particle-bound PPAHs were collected on quartz filters (Whatman, 150 mm), which were prebaked at 400°C for 5 hours to remove absorbed organic compounds. A high-volume sampler (HVS) (DH77, DIGITEL Elektronik AG, Hegnau, Switzerland)

equipped with a PM$_{2.5}$ impactor was used, operating at a flow rate of 500 L/min. After sampling, the collected filters were refrigerated at a temperature below 4˚C until extraction. The extraction and analysis of the 16 PPAHs followed the EPA TO-13A method with some modifications.

Prior to extraction, the collected filters are half-cut. A half filter is spiked with 1μg/mL of deuterated PAH standard, then extracted in 10mL of Toluene/Hexane (4:1v/v) by ultra-sonication for 90 min at 40˚C. The extract is to be concentrated into 1mL under Nitrogen (N$_2$) flow fixed at 10 L/min. After that, the concentrate is cleaned up on a HyperSep Silica SPE (Solid Phase Extraction) cartridge with 10 mL of hexane as an elution solvent. The sample was re-concentrated to 150 μL again under the same flow of N$_2$. The sample is then injected into GC-MS for analysis.

The PPAHs were analyzed with Thermo-Finnigan Trace GC-Ultra Polaris ITQ 900 MS coupled with AS 3000 II auto-sampler. Chromatographic separation was carried out on an Rxi-17Sil MS column (30 m × 0.25 μm film thickness× 0.25 mm film ID) with 99.999% purity Helium as carrier gas in a constant flow rate of 1 mL/min. The injection mode was splitless and set at 280 ˚C. The GC oven temperature was programmed from 80˚C (hold for 3 minute) to 170˚C (10˚C/min, hold for 1 min), to 180˚C (3˚C /min, hold for 0 min), to 270˚C (10˚C/min, hold time 0 min), then ramped to 300˚C (3˚C/min, hold for 10 min). The mass spectrometer was operated in full scan mode (50–350). The ion source temperature was 250˚C in electron impact mode (70 eV). The analytes were identified by their mass spectrum in which PPAHs and IS have a relatively intense molecular ion (mass-to-charge ratio (m/z)).

All the quality control (QC) and quality assurance (QA) were fulfilled. Limit of detection (LOD) and limit of quantification (LOQ) analysis was carried out using seven replicate extractions of a solution of the 16 PAHs at 0.1 μg/mL spiked with 1 μg/mL of deuterated standard. The results showed that ranges of LOD and LOQ limits of the 16 PAHs are 0.01–0.05 μg/mL and 0.03–0.16 μg/mL respectively. The results of seven replicate standards for two concentrations (0.1 and 4 μg/mL) revealed % RSD (percent relative standard deviation) less than 7%. The overall recovery of each PAH ranged between 90 and 108%. The deuterated standard calibration method was used to quantify the PAHs where it was found to be linear for the whole examined range of the 16 PAHs with a correlation coefficient >0.996. Concentrations of PAHs in the blanks were below the method detection limits during the analysis period.

## Source apportionment

Source apportionment was carried out with the ME-2 engine version of positive matrix factorization PMF [21] using source finder (SoFi) software toolkit for IGOR PRO [22]. The methodology of PMF was previously detailed and explained [23]. It is, in principle, a receptor model built on a mathematical equation that correlates different sets of elemental tracers (fingerprints) of distinct sources to identify and quantify their contributions to a set of tested samples. It explains the data set variability with a linear combination of factor profiles (sources) ($f_{i,k}$) representing the chemical tracers (PAHs) and their varying contribution ($g_{k,j}$). The resultant residual matrix is termed as ($e_{i,j}$). The input matrix is represented by ($x_{i,j}$). The indices $i$ and $j$ represent the number of samples and the number of chemical species respectively. The number of sources is represented by ($p$) and is determined and adjusted manually depending on the expectation from the sampling sites and their realistic environmental interpretation.

In this case, the equation below is explained by the input matrix being our PPAHs complete data set (from the three sites), and the result is the factor profiles (($f_{i,k}$)), the time series ($g_{k,j}$),

and a residual matrix ($e_{i,j}$).

$$x_{i,j} = \sum_{k=1}^{p} \left( f_{i,k} \cdot g_{k,j} \right) + e_{i,j}$$

### Cancer risk calculation

In this study, the carcinogenic potency of the measured PPAHs was determined using the widely applied "toxic equivalence factor" (TEF) method [24–29]. This approach which is based on the procedures of the Office of Environmental Health Hazards Assessment of California Environmental Protection Agency (OEHHA-CalEPA), accounts for the toxicity of all PAHs in the form of BaP equivalence [28].

$$\sum BaPeq = \sum (C_i \times TEF_i) \tag{1}$$

Where $\Sigma BaPeq$ is the total BaP equivalent concentrations, $C_i$ is the concentration of each PAH in ng/m$^3$, and TEFi is the toxicity equivalent factor of each PAH [28–30].

To evaluate the potential risk of developing cancer through inhalation and exposure to particle-bound PAHs, the OEHHA procedure was utilized. This established approach allows for the calculation of the lifetime excess cancer risk using the following equation:

$$Excess\ Cancer\ Risk(ECR) = \sum BaPeq \times UR_{BaP} \tag{2}$$

The equation takes into consideration the $UR_{BaP}$ value, which represents the number of individuals at risk of developing cancer from inhaling a BaP equivalent concentration of 1 ng/m$^3$ over a 70-year lifetime [31, 32]. The value of $UR_{BaP}$ utilized in this study was $1.1 \times 10^{-6}$ (equivalent to 0.11 cases per 100,000 people), as determined by the OEHHA of the CalEPA.

## Results

### Emissions of PM$_{2.5}$ at the three sites in Beirut

(Fig 2) shows the PM$_{2.5}$ box plot that represents the means and standard deviations of PM$_{2.5}$ levels at the three sites (referred to as BCD, AUB, and NSMU). The mean PM$_{2.5}$ level at BCD was found to be 18.3 μg/m$^3$, with a standard deviation of 6.8 μg/m$^3$. The maximum recorded

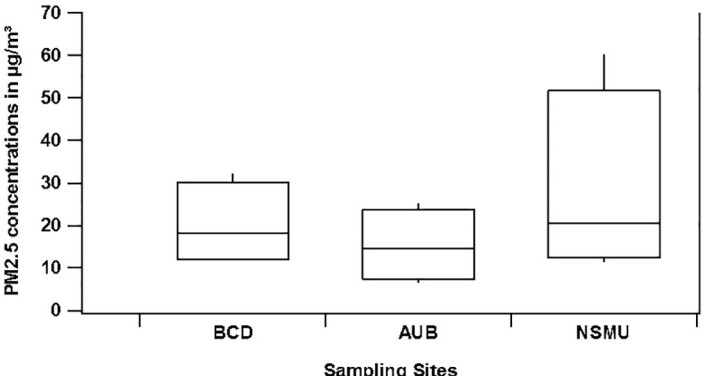

**Fig 2. Box plots of PM2.5 concentrations measured at BCD, AUB and NSMU.** Whisker bottom, box bottom, box top and whisker top represent 10, 25, 75 and 100 concentration percentiles.

value was 32.1 μg/m$^3$ measured on April 2022, while the minimum was 11.4 μg/m$^3$ in March 2022. Similarly, AUB site exhibited a mean PM$_{2.5}$ level of 14.3 μg/m$^3$, accompanied by a standard deviation of 6.0 μg/m$^3$. The highest and lowest values observed and measured in February 2023 at AUB were 25.7 μg/m$^3$ and 2.4 μg/m$^3$, respectively. NSMU displayed an average PM$_{2.5}$ level of 22.6 μg/m$^3$, with a relatively higher standard deviation of 13.7 μg/m$^3$. The highest and lowest values recorded at this site were 60.3 μg/m$^3$ in April 2023 and 8.3 μg/m$^3$ in May 2023, respectively.

The annual PM$_{2.5}$ level in Beirut of 17.19 μg/m$^3$, was found to exceed the recommended WHO value (5 μg/m$^3$) [33] by 243.8%. In addition, across all sites, 52% of all PM$_{2.5}$ measurements exceeded the daily limit. Specifically, at BCD, 60% of the data points exceeded 15 μg/m$^3$, indicating a prolonged period of elevated PM$_{2.5}$ levels. AUB site exhibited a lower percentage, with 39.3% of measurements surpassing the daily limit. In contrast, Site of NSMU demonstrated the highest proportion, with 75% of measurements exceeding 15 μg/m$^3$, highlighting a more significant concern for air quality deterioration.

## The concentration of particle bound polycyclic aromatic hydrocarbons (PPAHs) at the three sites in Beirut

The evaluation of PPAH concentrations at three different sites in Beirut provided insights into the temporal fluctuations, spatial variability, and potential carcinogenic effects of these compounds. As showed in (Table 1), for the BCD site (n = 13), the total 16 EPA PPAHs average was 20.52±11.27 ng/m$^3$; the highest concentration was 46.84 ng/m$^3$ (observed on April 2, 2022), and the lowest concentration was 8.92 ng/m$^3$ (observed on April 20, 2022). Similarly, in AUB (n = 17), the total 16 EPA PPAHs average concentration was 18.17±9.57 ng/m$^3$, the highest concentration was 36.38 ng/m$^3$ (observed on November 29, 2022), and the lowest concentration was 8.56 ng/m$^3$ (observed on February 28, 2023). Furthermore, in NSMU (n = 13), the total 16 EPA PPAHs average was 23.36±12.72 ng/m$^3$, the highest concentration was 49.93 ng/

**Table 1. Averages (Avg) and standard deviations (std), in ng/m$^3$, of individual and total 16 PPAHs in the three Beirut sites during the campaign.**

| The three sites in Beirut | BCD | | AUB | | NSMU | |
|---|---|---|---|---|---|---|
| PPAHs (Abreviation) | Avg | Std | Avg | Std | Avg | Std |
| Naphtalene (Nap) | 0.75 | 0.17 | 0.75 | 0.12 | 1.81 | 1.27 |
| Acenaphtylene (Acy) | 0.38 | 0.08 | 0.37 | 0.06 | 0.70 | 0.44 |
| Acenaphtene (Ace) | 0.42 | 0.21 | 0.55 | 0.61 | 0.22 | 0.12 |
| Fluorene (Flu) | 0.60 | 0.33 | 0.43 | 0.02 | 0.57 | 0.30 |
| Phenanthrene (Phe) | 0.89 | 0.42 | 0.73 | 0.19 | 0.99 | 0.40 |
| Anthracene (Ant) | 0.76 | 0.28 | 0.78 | 0.11 | 0.86 | 0.30 |
| Fluoranthene (Flt) | 1.62 | 1.18 | 1.32 | 0.60 | 1.33 | 0.74 |
| Pyrene (Pyr) | 2.20 | 1.37 | 1.71 | 0.92 | 2.25 | 1.34 |
| benzo[a]anthracene (BaA) | 1.98 | 1.12 | 1.35 | 0.93 | 1.60 | 1.15 |
| Chrysene (Chr) | 3.16 | 1.69 | 2.50 | 1.66 | 2.41 | 1.58 |
| Benzo[k]fluoranthene (BkF) | 3.25 | 2.43 | 3.34 | 2.49 | 3.85 | 3.79 |
| Benzo[b]fluoranthene (BbF) | 1.57 | 1.02 | 1.64 | 1.12 | 1.93 | 2.00 |
| Benzo[a]pyrene (BaP) | 2.41 | 2.07 | 2.24 | 1.71 | 3.26 | 3.88 |
| Benzo[g,h,i]perylene (BghIP) | 0.13 | 0.13 | 0.08 | 0.05 | 0.49 | 1.03 |
| Dibenz[a,h]anthracene (DahA) | 0.24 | 0.09 | 0.20 | 0.03 | 0.48 | 0.47 |
| Indeno[1,2,3-cd]pyrene (IP) | 0.16 | 0.11 | 0.17 | 0.10 | 0.59 | 0.64 |
| **Total 16 PPAHs** | **20.52** | 11.27 | **18.17** | 9.57 | **23.36** | 12.72 |

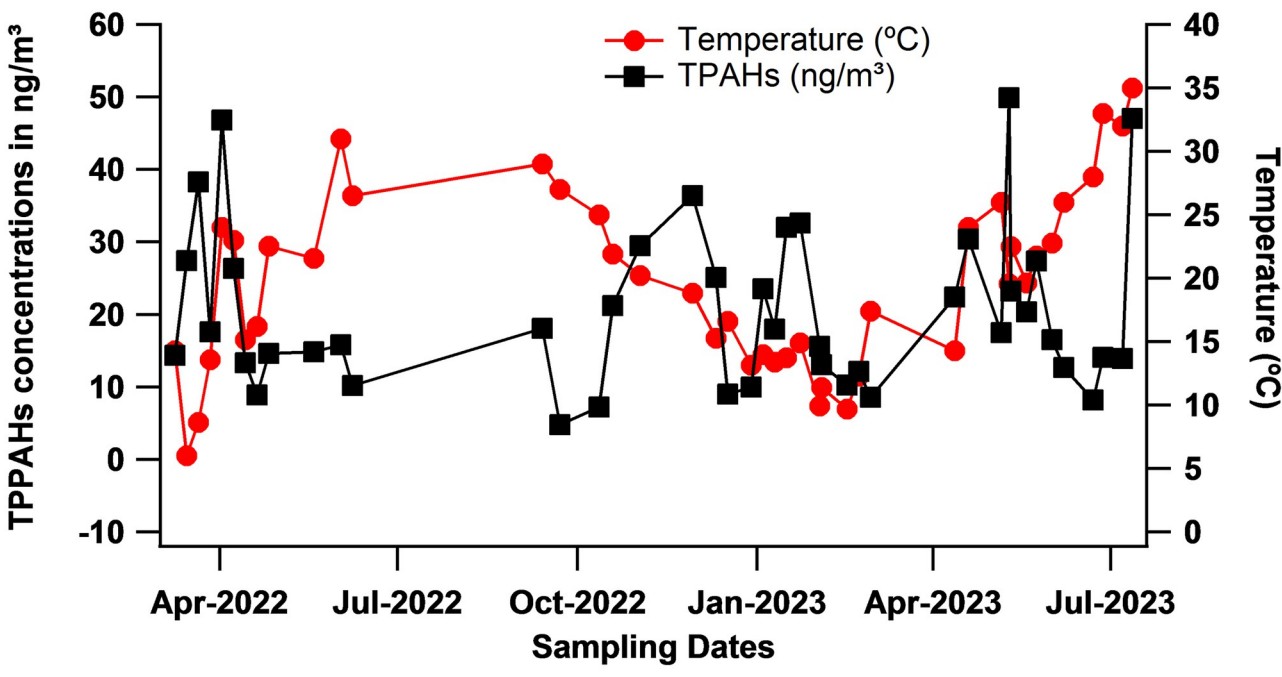

**Fig 3. Temporal variation of TPPAHs during the campaign started in March, 2022 and ended in July, 2023.**

$m^3$ (observed on May 10, 2023), and the lowest concentration was 8.20 ng/m$^3$ (observed on June 22, 2023).

As shown in (Fig 3), the measurements do not show a significant temporal variation in PPAHs concentrations throughout the year. In general the evaporation of PPAHs from the particulate to the gas phase and the prevalence of photodegradation reactions intensify during the summer and lead to a decrease in PPAHs concentrations in the particulate phase. However, in cities like Beirut, the demand for electricity is constantly increasing, driven by residential, commercial, and institutional activities such as schools, hospitals, and governmental buildings [34, 35]. This heightened demand is particularly pronounced during the hot summer season for cooling purposes. Consequently, this study found that PPAHs levels, including total PPAHs (TPPAHs), tend to be concentrated during summer as well as in winter and fall.

After conducting T.test, the data revealed a significant difference between the air quality at AUB and BCD areas (p = 0.01), as well as between AUB and NSMU (p = 0.001), with a lesser disparity observed between BCD and NSMU (p = 0.07). At AUB, air quality notably differs from that of BCD and NSMU. This variance may be attributed to factors such as the lower density of diesel generators and their emissions, with only one diesel generator at AUB compared to at least five in each BCD and NSMU site. Additionally, AUB's proximity to relatively tranquil streets and verdant surroundings likely contribute to its improved air quality compared to the more densely populated areas of BCD and NSMU. Conversely, BCD areas typically host a higher number of diesel generators with greater capacities operating for longer durations and surrounded by highway with moderate to low load of traffics. Similarly, NSMU exhibits fewer green spaces and a higher concentration of diesel generators relative to AUB and higher load of traffics, which contributes to its distinct air quality profile.

Of particular concern is the yearly average of BaP (benzo[a]pyrene), a well-known carcinogenic compound, which exceeded the annual guideline set by the European Union Directive 2004/107/EC, surpassing the threshold of 1 ng/m$^3$ with an average concentration of 2.6 ng/m$^3$.

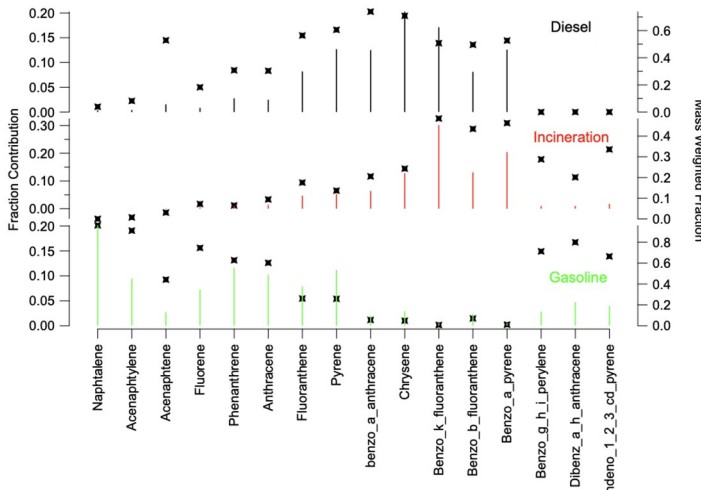

**Fig 4. The three factor profiles for factors 1,2 and 3 determined by PMF and the fraction of the species in each.**

## Source apportionment using Positive Matrix Factorization (PMF)

The positive matrix factorization (PMF) was used identify the sources of air pollution at the three Beirut sites based on the PPAHs profile. A three-factor model, linked to PPAHs, provided the best fit in the PMF analysis. Based on these references, three factors were identified. (Fig 4) illustrated the contribution of each factor at the respective sites.

Factor 1 predominantly featured medium molecular weight compounds (MMW) such as Fluoranthene, Pyrene, Benzo[a]anthracene, and Chrysene, alongside Benzo[k]fluoranthene [15, 36–45], indicative of emissions from diesel engines and generators. It is quite unusual to see diesel trucks making their way near the three specific sites. Factor 2 was characterized by high levels of Benzo[k]fluoranthene, Benzo[b]fluoranthene and Benzo[a]pyrene, consistent with incinerator emissions [16, 40], particularly medical ones. Factor 3 exhibited high loadings of Dibenz[a,h]anthracene and Indeno[1,2,3-cd]pyrene, with moderate loading of benzo[ghi] perylene [36, 41, 46–49] alongside elevated levels of low molecular weight PPAHs indicative of gasoline emissions. The already aged vehicle fleet determined to have an average of 18 years of age in 2017 [50], worsened by the economic downturn, further amplifies the concerning scenario, affirming substantial PPAHs emissions from vehicles. Gasoline-driven cars, dominant amidst Beirut's dense traffic, stand as the principal contributors to PPAHs in this context, exacerbating the city's heightened pollution levels. Based on the analysis of the factors contributions to each of the site a pie chart reflecting the distribution of source emissions is presented in (Fig 5).

Notably, Lebanon stood out among the few countries where incineration and diesel generators accounted between 67% and 81% of PPAHs emissions into the atmosphere. These findings underscore the significant impact of emissions from an old vehicle fleet, incineration mainly from hospitals, and the sprawl of diesel generators on the well-being of residents in Beirut.

## Excess Cancer Risk (ECR)

ECRs were calculated to assess the exposure risks based on the total PPAHs that were measured in this study. As shown in (Fig 6), the excess cancer risk for BCD, AUB and NSMU, exceeded the threshold for acceptable cancer risk set by EPA ($10^{-6}$) [51], as they were found to

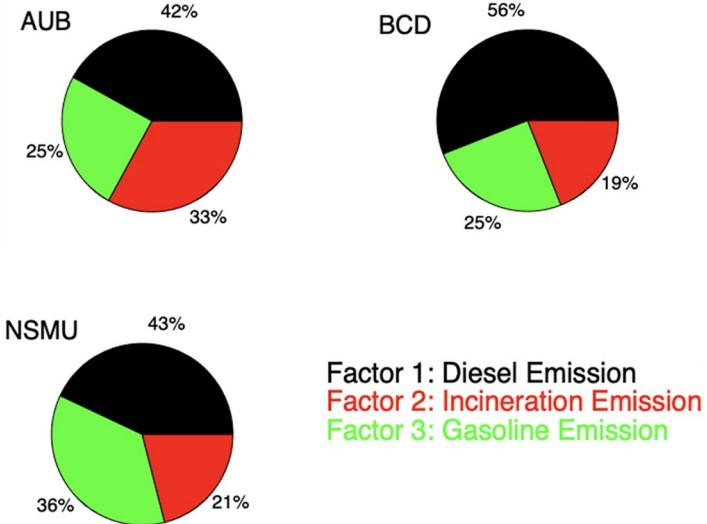

**Fig 5. The portion contribution of the three factors to the total ambient PPAHs.**

be $3.73 \times 10^{-6}$, $3.45 \times 10^{-6}$ and $5.05 \times 10^{-6}$, respectively. As expected, Benzo[a]pyrene had the highest contribution on the excess cancer risk in all the sampling sites (around 72%). Benzo[k]fluoranthene, Dibenz[a,h]anthracene, Benzo[b]fluoranthene, and Benzo[a]anthracene had also a great contribution on the excess cancer. Their corresponding percentage contribution to the excess cancer for BCD site were 10%, 7% 5% and 6%; for AUB site were 11%, 6% 5%, and 4% respectively, and for the NSMU site were 8%, 10%, 4% and 3%. Others measured PPAHs such as Indeno[1,2,3-cd] pyrene, Chrysene, Benzo[g,h,i]perylene, Pyrene, Fluoranthene, Naphtalene, Phenanthrene, Fluorene, Acenaphtylene, Acenaphtene, and Anthracene did not have a big contribution to the excess cancer risk (all together 2%). The excess cancer risk measured during the 2022–2023 period at the AUB site ($3.45 \times 10^{-6}$), is higher than previously measured values in the same site in 2016–2017 ($2.25 \times 10^{-6}$) [16] and in 2015 ($1.05 \times 10^{-6}$) [15].

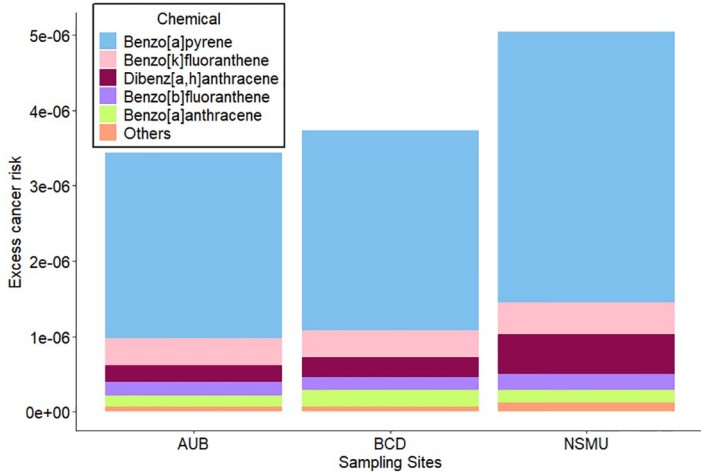

**Fig 6. Excess cancer risk of various species measured in each of the sampling locations.** Others correspond to the following species: Indeno[1,2,3-cd]pyrene, Chrysene, Benzo[g,h,i]perylene, Pyrene, Fluoranthene, Naphtalene, Phenanthrene, Fluorene, Acenaphtylene, Acenaphtene, and Anthracene.

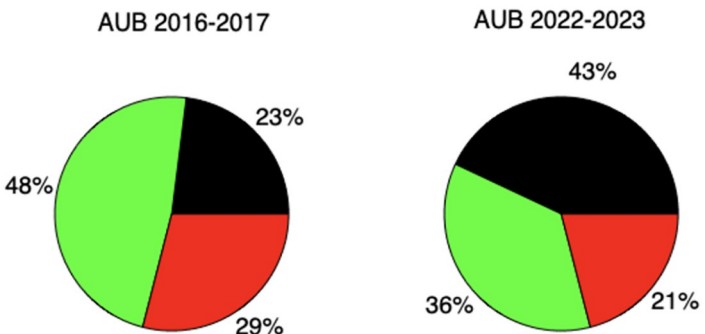

**Fig 7. Contribution of every factor in AUB before and after crisis.**

## Discussion

The evaluation of PPAHs concentrations at three separate locations in Beirut indicated notable variations over time and between sites, potentially posing carcinogenic risks. The average PPAHs levels in BCD, AUB, and NSMU exhibited significant fluctuations without specific values provided. These levels tended to be higher during periods of lower temperatures, aligning with previous research findings [13, 15, 16, 52]. The study found significant differences in air quality between AUB and BCD, AUB and NSMU, and to a lesser extent, between BCD and NSMU. AUB exhibited better air quality attributed to fewer diesel generators, tranquil surroundings, and green spaces compared to the greater number of diesel generators per building in BCD and to the more densely populated areas of NSMU. However, all sites showed concerning levels of BaP, a carcinogenic PPAHs compound.

A comparative analysis was conducted to assess the differences in the contributions of the three pollution factors at the AUB site between this study and the study conducted in 2016–2017. This analysis aimed to understand the impact of increased rationing hours on the AUB site. The contribution of diesel increased from 23% to 43%. In 2016–2017, diesel generators in Beirut operated for 3 hours per day. However, since 2021, they have been running for longer periods, ranging from 10–15 hours in 2021 to over 20 hours in 2022 and beyond, due to the electricity shortage caused by the grid shutdown (Fig 7).

Furthermore, there has been a notable decline in the proportion of emissions attributed to gasoline and incineration, dropping from 48% to 36% and 29% to 21%, respectively. This shift can be primarily attributed to the significant increase in operational hours of diesel generators, rising from 3 hours to 18 hours per day. Consequently, diesel combustion emissions have emerged as a more substantial contributor to pollution levels when compared to the findings of the earlier study conducted in 2017–2018.

Notable differences in the BaP concentrations were observed between this study and previous ones conducted at the same AUB site in 2015 [15] and 2016–2017 [16]. As shown in (Fig 8), there has been a 134% increase in BaP concentration since 2015. The significant differences in the data between 2022–2023 and 2015 ($p < 0.01$) and 2016–2017 ($p < 0.001$) provide strong evidence of the contribution of carcinogens from the extended operation hours of diesel generators.

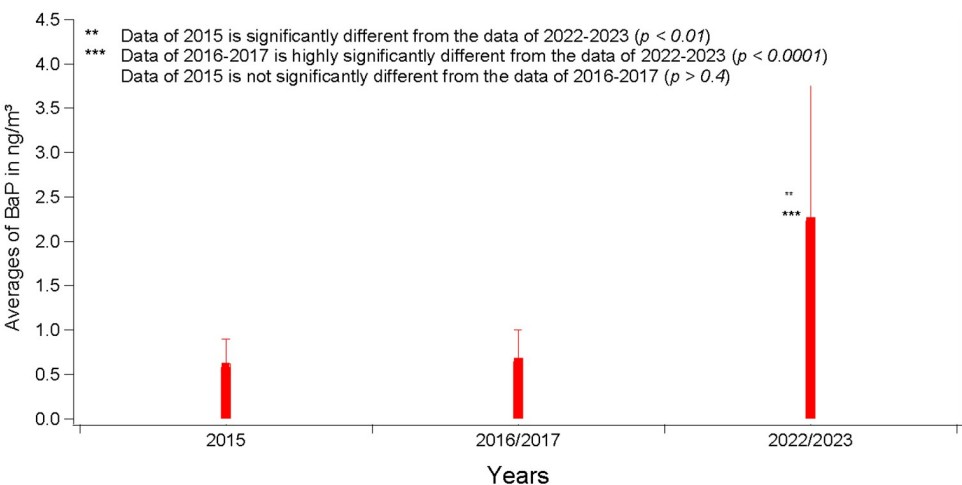

**Fig 8. The effect of the extension of diesel generators' operating hours on the levels of Benzo[a]pyrene in ng/m3 from 2015 till the period of this study's campaign in 2022–2023.**

Despite repeated recommendations and academic presentations, the local government has chosen to ignore these warnings. This has resulted in a sector of electricity producers and merchants operating outside the law, with no oversight from any government entity. The interventions from local authorities and international donors, rushing to help, have cost over $43 billion [53], with little to show for it. This situation highlights the urgent need for international action to curb emissions at the national level and for funding agencies to implement stringent accountability and auditing measures. Additionally, there is a critical need for these agencies to adopt sustainable, "carbon-free" funding strategies to support urban development in low- and middle-income countries (LMICs).

As the only legal national provider of electricity in the country, nearly every household in Lebanon is connected to the EDL grid. However, EDL's inability to meet the demands of Lebanon's residents constitutes a violation of their right to electricity. The high costs associated with services provided by diesel generator groups mean that electricity has become prohibitively expensive, forcing people to pay for this basic need at the expense of other necessities, thereby reinforcing the country's deep-seated inequality.

## Conclusion

Diesel generators were used to compensate for power shortages from Electricité du Liban (EDL). A year-long (2022–2023) comparative analysis at three Beirut locations (AUB, BCD, and NSMU) investigated the impact on 16 EPA particle-bound polycyclic aromatic hydrocarbons (PPAHs). The study revealed mean $PM_{2.5}$ levels of 18.3 $\mu g/m^3$ at BCD, 14.3 $\mu g/m^3$ at AUB, and 22.6 $\mu g/m^3$ at NSMU. Beirut's annual $PM_{2.5}$ level of 17.19 $\mu g/m^3$ exceeded the WHO recommended value by 243.8%, with 52% of measurements surpassing the daily limit. Average PPAHs concentrations were 20.52±11.27 $ng/m^3$ at BCD, 18.17±9.57 $ng/m^3$ at AUB, and 23.36±12.72 $ng/m^3$ at NSMU, with significant differences in air quality between the sites.

Of particular concern, the carcinogenic compound benzo[a]pyrene (BaP) exceeded the EU annual guideline of 1 $ng/m^3$. PMF analysis identified three pollution sources: diesel, incineration, and gasoline emissions, with diesel emissions predominating.

The elevated PPAHs levels result from inadequate regulation enforcement and unregulated sources, with a calculated twofold increase due to economic collapse and power plant

shutdowns between 2016 and 2023. This poses significant health risks to Lebanon's population and other regions affected by PAH transport. Effective mitigation and reduction strategies are lacking, highlighting the need for funding agencies and organizations to enforce higher accountability and promote sustainable "carbon-free" plans to support urban development in low- and middle-income countries (LMICs).

## Supporting information

**S1 File.**
(DOCX)

## Author Contributions

**Conceptualization:** Najat Saliba.

**Data curation:** Mohammad Al Hallak.

**Investigation:** Mohammad Al Hallak.

**Methodology:** Mohammad Al Hallak.

**Project administration:** Najat Saliba.

**Supervision:** Najat Saliba.

**Validation:** Mohammad Al Hallak.

**Writing – original draft:** Mohammad Al Hallak.

**Writing – review & editing:** Mohammad Al Hallak, Abdul Aziz Kurdieh, Amira Yassine, Rachel El Hage, Najat Saliba.

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
