## [Decision Letter · Decision Letter 0]

8 Aug 2024

PONE-D-24-28162

When do science recommendations stop being effective? The Case of the sprawl of diesel electricity generators in Beirut

PLOS ONE

Dear Dr. Saliba,

Thank you for submitting your manuscript to PLOS ONE. After careful consideration, we feel that it has merit but does not fully meet PLOS ONE’s publication criteria as it currently stands. Therefore, we invite you to submit a revised version of the manuscript that addresses the points raised during the review process.

We look forward to receiving your revised manuscript.

Kind regards,

Ranjit Gurav, Ph.D

Academic Editor

PLOS ONE

Journal Requirements:

5. We note that Figure 1 in your submission contain map/satellite images which may be copyrighted. All PLOS content is published under the Creative Commons Attribution License (CC BY 4.0), which means that the manuscript, images, and Supporting Information files will be freely available online, and any third party is permitted to access, download, copy, distribute, and use these materials in any way, even commercially, with proper attribution. For these reasons, we cannot publish previously copyrighted maps or satellite images created using proprietary data, such as Google software (Google Maps, Street View, and Earth). For more information, see our copyright guidelines: http://journals.plos.org/plosone/s/licenses-and-copyright.

Reviewers' comments:

Reviewer's Responses to Questions

**Comments to the Author**

1. Is the manuscript technically sound, and do the data support the conclusions?

Reviewer #1: Partly

Reviewer #2: Yes

2. Has the statistical analysis been performed appropriately and rigorously? 

Reviewer #1: Yes

Reviewer #2: Yes

3. Have the authors made all data underlying the findings in their manuscript fully available?

Reviewer #1: Yes

Reviewer #2: Yes

4. Is the manuscript presented in an intelligible fashion and written in standard English?

Reviewer #1: Yes

Reviewer #2: Yes

5. Review Comments to the Author

Reviewer #1: This manuscript presents a source apportionment study on a considerably long dataset at multiple sites in Beirut using polycyclic aromatic hydrocarbons as tracers. The study was performed by using positive matrix factorization applied to original data to evaluate the contribution of diesel electricity generators to polycyclic aromatic hydrocarbons. Furthermore, the excess cancer risk is calculated. The results provide interesting information about the sources of polycyclic aromatic hydrocarbons in Beirut. The manuscript is overall written in clear English, although the “Methods” and “Results” could benefit from a revision especially focused on the findings represented (see my specific comments below). I believe this manuscript is of interest to the scientific community and is coherent with the aims of this journal. Nevertheless, in my opinion, a few issues need to be addressed before the manuscript can be accepted for publication.

Specific comments

Lines 37 Please define the name of the sampling site when they are shown first.

Lines 183-189 The descriptions of source apportionment should be shown in detail. The current version is too simple to follow for the readers.

Lines 190-196 The descriptions of cancer risk calculation should be shown in detail. The current version is too simple for the readers to follow.

Lines 199. There is no figure 2.

Line 268. The quality of Figure 4 should be improved. The authors should present the source profiles for each factor.

Lines 273-286. There are no discussions for the source profile of each profile at all. The authors should compare the source profiles in this study with prior results in other studies. See the link https://doi.org/10.1016/j.envpol.2019.113046.

Lines 342-351. What are the energy policies recommended from the authors? Biomass has been regarded as a promising option for solid energy, which is promoted in the residential sector and firing power plants in some low-income and middle-income counties. The established programs and technologies of wood control burning requirements have been documented in the US, which contribute to the major efforts in regulating air quality in the Northeastern US where wood fuel is very important

See the link https://doi.org/10.1016/j.envpol.2023.123240.

EPA, 2020. The 2015 New Source Performance Standards for New Residential Wood Heaters. New Hydronic Heaters and Forced-Air Furnaces, NC, USA

Reviewer #2: General comment

The studies tackles an extremely important environmental issue, linking economic recession and fuel poverty with carcinogenic aerosol pollution, in a semi-quantitative manner. However, the approach taken is rather local, and the discussion is very limited in terms of comparisons with the international status. The source apportionment methodology should be explained in more detail, and the scope of PMF related analysis and valorization of data should be expanded. The manuscript must be revised before publication according to the specific comments listed below.

Specific comments

L58-60: Since the national provider was totally shut down, how is power provided during non-blackout periods? Are there secondary providers?

L66-67: What about the remaining 63%?

Introduction: The study highlights a very important subject, but the presentation is local in scope. The introduction should broaden the implications of this type of research by presenting similar impacts of fuel poverty in other areas of the world. Such an example is the severe air quality degradation in European cities due to the EU debt crisis that led to the increase of oil prices and the widespread use of biomass products for heating (see, for example, Kaskaoutis et al., 2021, Atmos. Environ.). Another is the use of extremely polluting, low-quality farming byproducts for cooking and heating purposes in low-income countries that rely on an agrarian economy (Saenz et al., 2021, Environ. Int.; Bhattu et al., 2024, Nat. Comm.). Include a brief introductory paragraph in this direction to place the study within an international context.

L122: Indicate the distance of the highway and whether it affects measurements.

L122-126: So, what is the type of this site? Still urban background but more central or traffic?

L129: Indicate the distance and traffic load of this road and whether they result in a “traffic” site characterization.

L131: It is unclear how this sampling strategy will help assess the generators' impact. This is the main feature of the paper and must be explained. In addition, the sample numbers per site/period must be provided.

L139: Did you use denuders or impregnated backup filters in the Chemcomb?

L176-177: It would be more meaningful if you expressed these values per m3 of sampled air (actual or nominal).

L182: So, did you perform any blank corrections?

L173-182: Was it possible to calculate the repeatability of your quantifications using replicate analyses? That would be useful for the PMF as well.

L183-189: The methodological presentation of the source apportionment is insufficient. All necessary details must be provided, including uncertainty calculations, model selection strategy, model evaluation, error evaluation, rotations, constraints etc. This section must be substantially expanded. It is also important to indicate if the PAH dataset from three sites was combined to perform the PMF analysis.

L195-196: Not enough. You should list the TEFs used and their bibliographical sources. Also, indicate the used CalEPA unit risk. You should also mention that BaPeq is calculated as an intermediate step.

L198: This section should discuss the factors i.e. meteorological, varying seasonal emissions, spatial source constrasts, to explain the differences observed between the three sites-periods. This is necessary because if the site characteristics are widely different there is no point in calculating annual mean PM2.5 or using a combined dataset in the PMF.

L217-218: What about the other two?

L235-236: However, there is substantial site variation per member. This should be discussed. Moreover, the heavy members appear to record the higher levels at MSNU. It should be hinted early why this is the case (traffic or diesel generators?).

L241-242: Specify that this is a “cooling” demand.

L247-249: What type of hypothesis testing is used? This should be relevant also to the sample sizes.

Table 1: Check “NSMU”.

L254-257: Indicate which is the relative effect of traffic at BCD and MSNU.

L262: The way the references are used in this section is problematic. You should indicate specific references for each of the three factors and discuss the ones supporting your characterization. Focus especially on PAH PMF studies in the Eastern Mediterranean and the Middle East. This is important for all three factors..

L275-277: There has been no indication up to now for this type of source affecting the data in the text, and it is unknown how it can impact the area and under which conditions. In general, you should link the factor contributions with wind data (e.g. are the sites downwind of the incinerators when high factor contributions are estimated?) and also consider the seasonal (spatial) variability of source contributions. The discussion at present does not support the factor characterization enough. For example, based on this profile, Factor 2 could also be related to biomass burning emissions.

L277-286: In general, gasoline cars tend to produce more heavy than light PAHs. Are there any diesel-powered trucks and other HDDVs in Beirut? It could be that this is a combined diesel-gasoline traffic factor. I would not consider naphthalene as a tracer, as its overwhelming majority is in the gas phase, and its particle fraction is extremely unstable; therefore, not suitable for PMF (same goes for Ace, Acy and probably Flu, Phe). Check also the seasonality of the factor.

L288-292: The source contributions should be calculated per site, and the spatial contrasts should be discussed. This also helps verify the sources. Consider this in the discussion.

L293: Why didn’t you calculate BaPeq and risks for each of the three sources? Using the source profiles, this is feasible. Consider this also in the discussion.

L314-324: These have already been discussed; take a more general approach.

L326-327: Indicate that you are comparing to a study sampling PM10, not PM2.5.

L342-346: Political statements that are not supported by references. You have to rephrase them in scientific language.

L346-347: Add reference.

L354-357: Very important conclusion, include it in the abstract.

6. PLOS authors have the option to publish the peer review history of their article (what does this mean?). If published, this will include your full peer review and any attached files.

Reviewer #1: No

Reviewer #2: No

---

## [Author Response · Author response to Decision Letter 0]

25 Sep 2024

1- Map was done using GIS thus it was solved by adding the source to the figure.

2- Permits, for doing field work, were represented in a word document with proofs (for transparency)

3- Methodology (lab protocol) was kept without trying to submit it independently.

4- We thank and appreciate the editor's work.

5- We thank and appreciate the reviewers' insightful comments and we trust that all questions and inquiries were posed effectively and with improved clarity.

---

## [Decision Letter · Decision Letter 1]

23 Oct 2024

When do science recommendations stop being effective? The Case of the sprawl of diesel electricity generators in Beirut

PONE-D-24-28162R1

Dear Dr. Saliba,

We’re pleased to inform you that your manuscript has been judged scientifically suitable for publication and will be formally accepted for publication once it meets all outstanding technical requirements.

Kind regards,

Ranjit Gurav, Ph.D

Academic Editor

PLOS ONE

---

## [Editor Report · Acceptance letter]

29 Oct 2024

PONE-D-24-28162R1 

PLOS ONE

Dear Dr. Saliba, 

I'm pleased to inform you that your manuscript has been deemed suitable for publication in PLOS ONE. Congratulations! Your manuscript is now being handed over to our production team.

Kind regards, 

on behalf of

Dr. Ranjit Gurav 

Academic Editor

PLOS ONE